# Role of the Virome in Vaccine-Induced Immunization

**DOI:** 10.3390/vaccines13090895

**Published:** 2025-08-23

**Authors:** Rossella Cianci, Mario Caldarelli, Paola Brani, Annalisa Bosi, Alessandra Ponti, Cristina Giaroni, Andreina Baj

**Affiliations:** 1Department of Medical and Surgical Sciences, Catholic University of Sacred Heart, 00168 Rome, Italy; mario.caldarelli01@icatt.it; 2Istituto di Ricerca e Cura a Carattere Scientifico (IRCCS), Fondazione Policlinico Universitario A. Gemelli, 00168 Rome, Italy; 3Department of Medicine and Technological Innovation, University of Insubria, 21100 Varese, Italy; pbrani@uninsubria.it (P.B.); annalisa.bosi@uninsubria.it (A.B.); aponti@uninsubria.it (A.P.); cristina.giaroni@uninsubria.it (C.G.); andreina.baj@uninsubria.it (A.B.); 4Laboratory of Microbiology, ASST Sette Laghi, 21100 Varese, Italy

**Keywords:** human virome, vaccine-induced immunization, immune response, Torque Teno Virus (TTV), herpesvirus

## Abstract

The human virome—comprising viruses that can persist in a host, those that benefit the host, and those that remain latent—has gained increasing acceptance as a modulator of immune response toward vaccination. The factors known to influence vaccine efficacy include host genetics, age, and bacterial microbiota, while the virome is a much less considered fourth dimension. This article reviews how components of the virome such as Torque Teno Virus (TTV), cytomegalovirus (CMV), Epstein–Barr virus (EBV), and bacteriophages impact both innate and adaptive immune responses, including mechanisms of immune pre-activation, trained immunity, and molecular mimicry from both beneficial and detrimental perspectives for vaccine-induced immunization. Emphasis is given to immunocompromised populations such as transplant recipients and those with HIV, where virome composition has been shown to correlate with vaccine responsiveness. Experimental models support clinical observations on how chronic viral exposures can either enhance or inhibit vaccine efficacy. Finally, we discuss virome-aware precision vaccinology and call for the integration of the virome in the development of immunization strategies, thus improving outcomes through customization.

## 1. Introduction

Vaccines elicit protective immune responses, but the magnitude and durability of this immunity can vary widely between individuals. While factors such as age, genetics, and the bacterial microbiome are known to impact vaccine efficacy [1], the human virome—the collection of viruses that persistently or latently infect us without causing acute disease—is an emerging piece of the puzzle. Many humans harbor a menagerie of commensal or latent viruses, including Torque Teno Virus (TTV) and various herpesviruses as part of their normal microbiota [2]. These chronic viral passengers, including bacteriophages, interact with the immune system continuously, shaping immune function in ways that can influence responses to vaccines. At present, over 200 viral species are known to infect humans and produce a variety of clinical manifestations, ranging from asymptomatic infections to severe and life-threatening diseases. However, a limited number of them—less than 30—have effective and approved vaccines available, emphasizing the importance of understanding host–virus interaction and the potential influence of the human virome on vaccine efficacy [3].

This review explores how the virome modulates vaccine-induced immunity. We integrate findings from human clinical studies and experimental models to examine both beneficial and detrimental effects of the virome. Key clinical contexts, such as vaccine responses in transplant recipients and immunocompromised hosts, are considered, alongside mechanistic insights into viral modulation of innate and adaptive immunity. Finally, we discuss future perspectives, including how virome profiling could enhance precision vaccinology.

## 2. Historical Overview: From the Microbiome to the Virome

Microbiome refers to the combined genomes of all microorganisms found in a specific environment, such as bacteria, archaea, fungi, and viruses. In 2001, Joshua Lederberg originally defined the microbiome, and since then, it has changed some fields of immunology by showing how the resident microbial communities shape immune development, regulate tolerance, and modulate inflammatory responses. With very few exceptions, the microbiota’s bacterial constituents have been under constant investigation, whereas the viral component—the virome by definition—has recently emerged under the spotlight, rather fortuitously, following the advent of high-throughput sequencing and metagenomic analyses [4,5].

The human virome encompasses all viruses associated with the human body, including eukaryotic viruses that infect human cells, bacteriophages that infect the bacterial microbiota, and endogenous viral elements embedded within the human genome. These viruses could be pathogenic, commensal, or latent, with viral activity often countering symptoms rather than causing them for a long time in some anatomical niches. Over 200 viruses have been characterized as being able to infect humans; however, few of these viruses are implicated in the diseases of immunocompromised individuals [6]. The persistence and distribution of virome members depend ultimately on tissue type, immune status, and the age of the host. The temporal development of the human virome, its major sources of acquisition, and its interplay with immune maturation are summarized in Figure 1.

Many viral families are associated with good health: anelloviruses, such as the Torque Teno Virus, are small circular ssDNA viruses found everywhere in blood and tissues, considered mostly non-pathogenic and sometimes used as surrogate markers of immune competence [7]. Herpesviruses, such as cytomegalovirus (CMV), Epstein–Barr virus (EBV), and human herpes virus 6 (HHV-6), can infect lymphocytes, epithelial cells, or neurons, remaining latent and then periodically reacting; these viruses are mainly immunomodulatory in nature, creating problems particularly for transplant recipients and aging populations [8]. The pathogenicity of polyomaviruses, such as the BK and JC viruses (BK/JC), which usually persist in renal and urinary tissues, causing no symptoms, emerges only when the individual is immunosuppressed. Human papillomaviruses (HPVs), for example, include both commensal and oncogenic strains. The redondoviruses are a new category of small circular DNA viruses that have been largely described as colonizing the oro-respiratory tract, where they may increase in numbers during inflammatory or critical illness [9]. On the contrary, CrAss-like phages dominate the gut virome, which targets species from the phylum Bacteroidetes and plays a significant role in regulating the composition and function of the intestinal microbiota [10].

The modes of replication and persistence vary from one virus to another. Herpesviruses reside latently in lymphoid or neural tissues; anelloviruses replicate in the peripheral blood; polyomaviruses are found in the renal epithelium; while bacteriophages are mainly found in the gut lumen and mucus layer. Redondoviruses are highly abundant in the upper respiratory tract. The human body is home to upwards of 10^12^–10^13^ virus-like particles in terms of abundance, with the densest virome reservoir being in the gut [6].

The opportunities for pathogenic activity for viruses exist on this spectrum. Some viruses are pathogenic, while others quietly persist, causing no harm to the host under conditions of homeostasis and maybe even contributing to immune education. Exposure to low levels of virus continuously stimulates innate immune receptors, altering cytokine production and influencing immune cell maturation, thus suggesting the virome is not simply an innocent bystander but, instead, is interacting in the modulation of immune tone and perhaps even vaccine responsiveness.

Knowledge about what viruses persist in which anatomical locations and how their presence interacts with the dynamics of the immune system will help in the appreciation of the fine-tuned mechanistic interplay between host, microbiota, and immune function. The more we learn about the virome, the clearer it is that it sets the tone for immune maturation and responses to immunological perturbations, such as vaccination.

A summary of the major viral families identified in the human virome, their genomic characteristics, anatomical niches, and classification as commensal or pathogenic is presented in Table 1.

While the virome is generally accepted as referring to those viruses with long histories of being residents, acute viral infections may allow some viruses, under certain conditions, to persist. For example, some RNA viruses have shown the ability to persist at low levels in immune-privileged sites or under situations of chronic immune activation. The presence of persistent viral RNA or proteins can contribute to long-term immune modulation, epigenetic alteration, and modified cytokine profiles, even if there is no complete replication or production of infectious progeny [12]. Hence, this raises the potential that viruses introduced through acute infections may later find a place within the host’s virome, particularly in the context of immunocompromised or inflamed areas. The immunological consequences of such forms of viral persistence remain poorly understood but may include sustained activation of the innate immune receptors, exhaustion of adaptive immune responses, or de facto immune tolerance. Viral persistence, either because of systematically latent viruses or of unresolved acute infections, represents an integral dynamic of the virome, potentially relevant for vaccine responses and immune homeostasis [13].

## 3. Virome-Mediated Modulation of Innate and Adaptive Immunity

The human body is host to a vast array of viruses, the majority of which do not manifest as overt disease [2]. These include eukaryotic viruses that establish persistent infections, bacteriophages that infect the human microbiome, and even endogenous retroviral elements within the human genome [14]. Collectively termed the virome, many of these viruses are considered commensal, persisting in the host without apparent harm [15]. It is noteworthy that only approximately half of the known human-infecting viruses are pathogens; the remainder are frequently asymptomatic or benign [15]. Anelloviruses, including TTV, are classified as non-pathogenic small circular DNA viruses. On the other hand, the herpesviruses, large DNA viruses, possess potential pathogenicity and are associated with a variety of human diseases, while also being capable of establishing latency. These resident viruses are not idle passengers but can actively engage and modulate the host immune system. Accumulating evidence indicates that commensal viruses can play beneficial roles in immune homeostasis. In animal models, chronic intestinal viruses have been shown to attenuate intestinal inflammation through sustained interferon-β production by plasmacytoid dendritic cells (pDCs) [16]. Nicoles et al. demonstrated that in mice, the presence of a natural enteric virome helps maintain intraepithelial lymphocytes through RIG-I-mediated IL-15 production. This study highlights that viral signals contribute to mucosal immune tone [17]. A seminal study found that latent herpesvirus infections in mice conferred protection against bacterial infections by stimulating basal immune activation (e.g., chronic γ-herpesvirus infection increased interferon-γ levels that helped mice resist *Listeria monocytogenes*) [18]. These findings lend support to the concept that the virome, akin to the bacterial microbiome, provides continuous stimuli that can “train” or tune the immune system. Commensal viruses can thus serve as a constant immune exercise, thereby enhancing the body’s readiness to respond to other threats. Indeed, another study demonstrated that colonizing germ-free mice with a persistent norovirus restored normal intestinal morphology and immune cell development, replacing missing bacterial stimuli [19].

Conversely, the virome’s influence can be double-edged. Persistent viruses have evolved immune evasion strategies to avoid clearance, and their presence can drive chronic immune activation or immune subset skewing. In the event of waning immune surveillance by the host, for instance, during immunosuppression, commensal viruses have the potential to replicate unchecked, thereby potentially exacerbating the severity of an underlying pathology [20]. Studies conducted on human immunodeficiency virus (HIV)-positive subjects have demonstrated that advanced HIV infection, characterized by low CD4^+^ T-cell counts, is accompanied by a substantial expansion of the enteric virome, including adenoviruses and anelloviruses. This expansion has been observed to correlate with intestinal inflammation and disease progression [20]. Consequently, a healthy immune system and the virome exist in a dynamic equilibrium. It is generally accepted that the immune system normally constrains latent viruses. In return, some viruses help stimulate and shape immunity. In the event of a shift in this equilibrium, whether due to immunodeficiency or immunosuppressive therapy, there is a potential for the virome to expand, thus contributing to chronic inflammation [21,22]. These observations underscore the notion that the virome is an integral component of the immune environment, capable of both bolstering and impairing immune function. It can thus be inferred that the virome composition of an individual may significantly influence the immune system response to a vaccine. The following sections delve into specific viruses and contexts to illustrate this impact.

Beyond their distinctive roles in immunity, virome components differ fundamentally in their scope of influence on the host organism. The modulation of the bacterial microbiome structure and activity by bacteriophage lysis of bacterial species and gene expression interactions, which can also contribute to horizontal gene transfer, shapes dynamics through microbial diversity and stability effects on digestion, metabolism, and mucosal immunity [23]. There are more studies associating dysbiosis of the virome with noninfectious diseases like type 2 diabetes, obesity, neurodevelopmental disorders (e.g., autism spectrum disorder), and some cancers. Many of these associations are observational and indicate that the virome has a significant role in not just modifying immune tone but broader systemic physiological processes [6,24,25].

Over the last years, vaccinology has facilitated the use of biological tools to study immune responses to vaccination [26]. Historically, considerable effort has been directed towards elucidating the bacterial microbiota’s ability to modify immunogenicity. However, recent studies have highlighted the human virome, defined as the community of viruses present in the human body, and its potential implication in the efficacy of vaccines [27,28].

The virome exerts a profound influence on the immune response to vaccines, encompassing both innate and adaptive immunity, in addition to epigenetic and indirect mechanisms that are sustained by the bacterial microbiota.

This concept has recently been the focus of debate, with particular attention being directed towards the potential involvement of the virome in modifying vaccine responses. This involves stimulation of innate immunity receptors, the development of cytokines such as IFN and IL-6, as well as IL-10, and the induction of trained immunity, especially in susceptible subjects [29].

The immune activation state is characterized by its chronic nature, manifesting as a low-grade and continuous process induced by several viruses residing within the virome. These include latent or persistent forms, such as cytomegalovirus or human Herpes virus 4. The process may involve a preliminary activation of different cell type, such as dendritic and natural killer cells, and macrophages, thereby enhancing their ability to respond to novel antigen presentation, for instance, vaccine antigens [30,31,32]. Furthermore, the activation of pattern recognition receptors such as Toll-like receptors, RIG-I-like receptors, and inflammasomes (e.g., NLRP6, NLRP9b) by viral constituents will induce the production of type I and type III interferons (IFNs). These are generally considered antiviral and tend to amplify adaptive responses and activate pre-cytokine environments in anticipation of the vaccine response [32,33].

The mechanism is illustrated in Figure 2, which shows the activation of innate immune system activation receptors (e.g., TLR-9 and RIG-I) by virome components, leading to the production of interferons, pro-inflammatory cytokines (e.g., IL-6, TNF) or regulatory cytokines (e.g., IL-10), which in turn modulate the immune response to the vaccine.

In mice, NLRP6 has been observed to engage DHX15-helicase upon binding to viral RNA. This interaction facilitates the binding of the NLRP6 inflammasome to the mitochondrial antiviral signaling protein complex, thereby triggering the release of IFN-α/β and IFN-γ. Consequently, this process serves to inhibit the replication of various viruses, including encephalomyocarditis and the norovirus [34]. Another mechanism involves NLRP9b binding to helicase DHX9, which facilitates the recognition of rotavirus RNA and the activation of pyrogenase. This process, in turn, induces pyroptosis, contributing to the removal of infected cells [35].

The virome also defines the activation and differentiation of T and B lymphocytes along the axis of adaptive immunity. It has been observed that viruses shift the activation direction towards the Th1 phenotype, thereby triggering pro-inflammatory cytokines, including IFN-γ. Conversely, other viruses have been observed to promote the generation of Tregs, which possess the capacity to suppress immune activity, thus exerting a negative effect on vaccine responses [32]. Similarly to T cells, B cells may also benefit from a previously activated immune environment, therefore promoting the maturation of antibodies, class response, and immunological memory.

This is another level of regulation, referred to as trained immunity. Consequently, chronic exposure to latent viruses is associated with epigenetic changes (for example, DNA methylation, histone modification) in myeloid progenitors, resulting in a continuous predisposition to respond more vigorously to subsequent stimuli, including vaccine antigens. This mechanism, while not antigen-specific, has the potential to enhance the responsiveness to subsequent vaccinations [36].

Finally, the virome also acts indirectly through modulation of the microbiota. For example, bacteriophages affect intestinal bacterial composition, as well as key metabolites, such as short-chain fatty acids (SCFAs), which are implicated in the maturation of regulatory T cells. Some phages have been observed to directly stimulate the production of pro-inflammatory cytokines, such as IFN-c, independently of bacteria, thus demonstrating of autonomous immunomodulatory potential [5].

A second mechanism is molecular mimicry, whereby certain viral proteins exhibit structural homology with the vaccine antigens, potentially generating cross-reactions. These cross-reactions could be advantageous, favoring the process of “pre-immunization”, or conversely, they could lead to the induction of tolerance or immune exhaustion [26].

Molecular mimicry has been demonstrated to elicit an immune response, which may either lead to anticipation and enhancement of immunization, or to the development of immune tolerance or lymphocyte exhaustion.

This phenomenon has been exemplified in African pediatric populations, where exposure to some enteric viruses has been linked to impaired responses to oral rotavirus vaccines [37].

Li et al. have dealt with this mimicry between the commensal viruses and the exogenous antigens that can change the antigenic presentation by dendritic cells, thus directing the activation of lymphocytes to the regulatory phenotypes [32].

Tolerogenic activation can also be compartmentalized at the mucosal level, as noted by Cadwell et al., which would explain the differences in the efficacy of the same vaccine in different individuals, depending on their resident virome [38].

Several recent studies have indicated that prior exposure to certain viruses can epigenetically reprogram cells of innate immunity, termed “trained immunity”. Nonspecific memory may elicit more rapid or even, in certain instances, subdued responses to future stimuli, which may include vaccines [36].

An illustration of the viroma’s effect on vaccination has been shown by a longitudinal study on 122 Ghanaian children vaccinated with Rotarix (RVV), which shows evidence of a negative correlation between gut bacteriophage diversity and post-vaccine seroconversion. The presence of Enterovirus B and cosaviruses in some subjects was also associated with a reduced immune response [37].

It is important to note that phages (bacteria viruses) can compromise the efficacy of vaccines. This can occur either through changes in the composition of the microbiota or through direct modulation of the immune system, for example, by stimulating IFN-γ production even in the absence of bacteria [39].

Several reports suggest that the virome, particularly the intestinal virome, may represent an important immunomodulatory factor concerning vaccine responses. In certain conditions, it enhances the immunogenicity, while in others, it has been observed to diminish it, as evidenced by data on subjects infected with SARS-CoV-2 that have exhibited significant alterations in their intestinal viral composition [5,40,41].

Molecular mimicry is another strategy through which the virome can manipulate immune responses to vaccines. Some proteins present in the viral agents are structurally like the vaccine antigens or those of the host, hence leading the immune system into cross-reactive responses, which may be beneficial, for example, by training the immune system before vaccination, or detrimental, by leading to immune tolerance or functional exhaustion [42]. For example, studies indicate that enteric viruses, such as the Enterovirus B and Cosavirus, potentially inhibit responses to oral rotavirus vaccines by epitopically mimicking them, hence reducing seroconversion. The extent of this is still unknown, but considerations of molecular mimicry in virome-aware vaccine design should be considered for populations that are normally exposed to a great deal of virome variety [42].

Such dynamics are built into plans for personalized vaccine strategies regarding individual virome composition and virome–microbiota–immune system interactions. Before examining the role of major viroma viruses in the vaccine response, Table 2 summarizes their essential characteristics and potential immunological impact.

Evidence from both observational and experimental studies suggests that an individual’s virome composition likely influences both the quantity and quality of vaccine-induced immune responses. Some viruses, such as CMV and EBV, are considered persistent in the host and alter immune memory, thereby promoting the expansion of T-cell subsets and subverting cytokine profiles through chronic stimulation with antigens [44]. Commensal viruses, such as anelloviruses and gut-resident bacteriophages, can modulate innate immunity by engaging pattern-recognition receptors, thus impacting the threshold of vaccine-induced activation [10,58]. Active or recently resolved viral infections may compete for immune resources, thereby siphoning cellular and humoral responses away from vaccine antigens [45]. On the contrary, certain pre-existing viral exposures may act as adjuvants, facilitating responses via trained immunity or cross-reactive T- and B-cell memory [59]. Together, these observations underpin the hypothesis that the human virome represents a dynamic and individualized immunologic context that can either promote or dampen vaccine efficacy, depending on its composition, activity, and interaction with host immunity.

The hypothesized mechanisms through which the virome may modulate vaccine-induced immunity are illustrated in Figure 3.

## 4. Torque Teno Virus: A Sentinel of Immune Status

TTV, a ubiquitous member of the Anelloviridae family, is one of the most abundant components of the human virome, with infection rates exceeding 90% globally and often acquired early in life [60]. A schematic representation of TTV biology, its known immunomodulatory pathways, and potential clinical implications for vaccine responsiveness is shown in Figure 4.

Although traditionally considered apathogenic, TTV is now gaining attention as a non-invasive biomarker of immune competence—a so-called “gentle spy” of the immune system [46]. In immunologically healthy individuals, TTV plasma levels are typically low (<10^4^ copies/mL) [61]. In contrast, states of immunosuppression, such as post-transplant immunosuppression or untreated HIV infection, are associated with marked increases in TTV viremia [48]. This inverse relationship suggests that an intact immune system constrains viral replication, whereas immune compromise allows TTV to proliferate. Elevated TTV levels have also been linked to systemic immune activation and inflammation, indicating that TTV may contribute to immunological “background noise” in these settings [62].

Beyond passive monitoring, TTV appears to interact directly with immune pathways. The viral ORF2 protein has been shown to suppress NF-κB signaling, dampening pro-inflammatory cytokines like IL-6 and IL-8 [47]. Meanwhile, its CpG-rich DNA can activate Toll-like receptor 9 (TLR9) in plasmacytoid dendritic cells and other immune cells [54], inducing interferon responses. Intriguingly, TTV carries both stimulatory and inhibitory CpG motifs, varying by strain, suggesting a capacity to fine-tune innate immune sensing [47,50]. In addition, TTV-encoded microRNAs may modulate antiviral responses, including interferon signaling [63]. These findings highlight a dual role for TTV: it may suppress certain immune signals while stimulating others, with potential implications for host defense and immune readiness.

This dynamic interaction naturally raises questions about TTV’s influence on vaccine-induced immunity. Growing clinical evidence suggests that high TTV loads correlate with poor vaccine responsiveness, particularly in immunosuppressed populations. For example, kidney transplant recipients with TTV levels above ~10^6^ copies/mL prior to SARS-CoV-2 mRNA vaccination were significantly less likely to seroconvert [64]. Similar associations were reported in lung transplant recipients, many of whom remained poor responders despite multiple booster doses [65]. A multi-center study further confirmed that high TTV titers at the time of vaccination predicted blunted humoral and cellular responses to SARS-CoV-2 vaccines [66]. Importantly, while TTV may not directly impair vaccine responses, its plasma load serves as a robust surrogate for immune competence—a real-time indicator of who may or may not respond adequately to immunization [61]. However, a causal role cannot be excluded, particularly if TTV’s NF-κB suppression directly interferes with vaccine immunogenicity. This pattern extends beyond COVID-19 vaccines. In recipients of influenza or hepatitis B vaccines, lower TTV levels consistently predict better immune responses, reinforcing its potential role as a universal biomarker in vaccinology [22,51]. Clinically, TTV monitoring could help stratify patients for tailored immunization approaches, including high-dose formulations, additional boosters, or immune-enhancing adjuncts. Investigational strategies—such as modifying immunosuppressive regimens or reducing TTV via antivirals—are being explored to determine whether improving immune status can restore vaccine efficacy. Ultimately, while TTV is not a pathogen in the conventional sense, it exemplifies how the commensal virome may shape the immune landscape, influencing how the host responds to vaccination and immune challenge.

## 5. Latent Herpesviruses and Immune Modulation of Vaccination

Latent herpesviruses, including cytomegalovirus (CMV), Epstein–Barr virus (EBV), and human herpesvirus 6 (HHV6), constitute a stable and influential part of the human virome. A schematic overview of latent human herpesviruses, their sites of persistence, immune evasion strategies, and potential implications for vaccine responses is provided in Figure 5.

Unlike TTV, herpesviruses are not considered harmless: they can cause overt disease under certain circumstances, particularly in immunocompromised individuals. Even in immunocompetent hosts, however, they persist subclinically, and continuously shape immune architecture. CMV, in particular, has emerged as a major driver of immune remodeling and immunosenescence [52].

By early adulthood, approximately 50–70% of individuals are CMV-seropositive. Over time, CMV induces a significant expansion of CMV-specific T cells, which may comprise up to 30% of memory CD4^+^ T cells and up to 50% of memory CD8^+^ T cells [67]. These populations often exhibit a late-differentiated, “senescent” phenotype, marked by CD57 expression, loss of CD28, and telomere shortening, suggestive of repeated antigen stimulation and reduced functional flexibility [68]. CMV infection is also associated with a decreased CD4^+^:CD8^+^ ratio and persistent low-grade inflammation [53], hallmarks of an aging immune system. EBV, while primarily infecting B cells, also alters immune balance by establishing latency in memory B cells, and has been implicated in autoimmune and lymphoproliferative disorders. These chronic alterations in immune structure raise critical questions about how latent herpesvirus infections affect responses to vaccination.

Most data have focused on CMV, especially in the context of vaccines for the elderly. Studies consistently demonstrate that CMV-seropositive older adults tend to mount weaker humoral responses to flu vaccines than CMV-negative counterparts [52,67]. A meta-analysis by van den Berg et al. showed a trend toward reduced influenza vaccine responsiveness in CMV-positive individuals, although not always statistically significant [69]. In contrast, younger adults under 60 appear largely unaffected by CMV status, suggesting that viral influence becomes more apparent in the context of aging or immune decline.

Beyond influenza, CMV-seropositivity has been associated with impaired responses to other vaccines. In young adults receiving experimental Ebola vaccines, CMV-seropositive individuals showed significantly lower glycoprotein-specific antibody titers and a higher proportion of terminally differentiated T cells (CD57^+^KLRG1^+^), across geographically distinct cohorts in the UK and West Africa [54]. Similarly, in patients with chronic kidney disease, CMV status predicted poor antibody responses to the 23-valent pneumococcal polysaccharide vaccine (PPV23), even more strongly than renal dysfunction itself [56]. Interestingly, one study in patients with type 2 diabetes found an opposite trend, where CMV-seropositive individuals had better responses to the influenza vaccine [57].

This paradox suggests that CMV’s effects may depend on the underlying inflammatory or metabolic milieu.

Some have speculated that, under certain contexts, CMV’s immune stimulation could function as an adjuvant, whereas in other contexts, it may predominate as an immune sink that diverts resources. This hypothesis still remains to be explored [70].

Interventional evidence further strengthens the case for a causal role. A 2022 clinical trial in CMV-positive patients with ANCA-associated vasculitis showed that those receiving antiviral prophylaxis (valacyclovir) had fewer CMV reactivations and mounted stronger immune responses to a pneumococcal conjugate vaccine (PCV13) compared to a placebo [71]. This trial supports the hypothesis that controlling subclinical viral activity may restore immunocompetence and improve vaccine efficacy in vulnerable groups.

The role of EBV in modulating vaccine response is less well understood. While EBV latency alters the B-cell compartment and is associated with chronic immune activation, direct evidence linking EBV to poor vaccine outcomes remains limited. However, in co-infection settings—such as CMV + EBV or EBV + HIV—interactions may exacerbate immune dysfunction, particularly in transplant patients or infants with congenital infections [68]. Indeed, early-life CMV infection has been linked to impaired vaccine responses in infancy, including to measles and polio vaccines [72,73]. This effect is compounded in HIV-infected infants: studies in Zambia showed significantly lower polio antibody titers in co-infected children compared to uninfected peers [74]. In people living with HIV (PLWH), CMV co-infection is nearly universal and contributes to persistent immune activation, microbial translocation, and expanded CD8^+^ T-cell compartments—all of which correlate with suboptimal vaccine outcomes [68].

Insights suggest several pathways through which CMV may impair vaccine responses. These include the following: (1) ”Immune space” competition, where CMV-specific T-cell clones dominate the repertoire, reducing the availability of naïve T cells for novel antigens [75]; (2) immune exhaustion, as persistent CMV antigen drives terminal differentiation and limits the functional plasticity of T and NK cells [76,77]; (3) interference with innate signaling, since CMV encodes proteins that downregulate MHC molecules and suppress interferon pathways, impairing the adjuvant effect of vaccines [78]; (4) dendritic cell dysfunction, potentially due to chronic reactivation of CMV or similar viruses that desensitize pattern recognition receptors and skew APCs toward suppressive profiles.

In summary, latent herpesviruses—especially CMV—act as subtle yet powerful modulators of immune readiness. Their effects may vary depending on host age, immune status, comorbidities, and the specific vaccine in question. While in some contexts CMV-induced immune stimulation may paradoxically enhance vaccine responses, the predominant effect appears to be immunological “distraction” or exhaustion. Emerging data suggest that targeted virome modulation—through antivirals or future vaccines—could become a viable strategy to enhance vaccine efficacy in immunologically vulnerable populations.

## 6. Other Commensal Viruses and Vaccine Responses

Although TTV and CMV have attracted significant research regarding their potential role in immune modulation and vaccine responsiveness, they still represent only a small portion of the human virome. This virome may also play a crucial role in modulating immune responses following immunization. Several other chronic or persistent viral infections, mostly neglected, may also meaningfully impact vaccine efficacy.

Polyomaviruses, such as the BK/JC, are known to remain latent in renal and other tissues. In immunocompromised hosts, especially organ transplant recipients, BK virus reactivation is common and could lead to complications such as polyomavirus-associated nephropathy [43]. Our understanding of how these viruses affect vaccine responses is limited, but an increased polyomavirus burden may also indirectly reflect states of profound immunosuppression, akin to TTV [58]. A growing number of studies on the response to vaccines point to these viruses since they may serve as indicators of immune status.

Chronic hepatitis B and C are states of continuous immune activation and dysfunction, especially within the hepatic microenvironment. For example, untreated hepatitis C infection is often associated with increased production of inflammatory cytokines and immune exhaustion features [79]. While many patients with these infections remain capable of developing good vaccine responses, especially after antiviral therapy clears the virus, chronic liver viral infections are an example of how actively replicating members of the virome can modulate host immunity, with a mechanism akin to that seen in HIV/AIDS or hepatic decompensation [80].

An interesting example is that of Human Pegivirus (HPgV). This RNA virus is usually non-pathogenic, but it can alter immune responses in co-infected individuals. In HIV-infected patients, HPgV infection appears to slow disease progression and improve survival, likely through immunomodulation [81,82]. Though the speculation on its effects on vaccine response is abundant, the findings would suggest that some virome features may be involved in beneficial immune modulation rather than adverse effects.

Aside from those already mentioned, the gut virome represents an enormously large and dynamic ecosystem, primarily populated by bacteriophages, and sporadically interspersed with persistent eukaryotic viruses such as enteroviruses or noroviruses. Bacteriophages may indirectly affect immune homeostasis by modulating the composition of the bacterial microbiota [83]. Disruption of this phage–bacteria equilibrium (e.g., lytic-phage blooms) may destabilize the gut ecosystem, conferring immune perturbation that may negatively impact the efficacy of mucosal or oral vaccines. This is a valid consideration for vaccines, such as the oral polio or rotavirus, for which differences in efficacy across geographical regions have been attributed in part to differing enteric virome diversities [84,85]. Furthermore, asymptomatic enteric infections can provide chronic immune stimulation, either promoting vaccine responses through interferon priming or suppressing them via immune regulatory pathways.

Similarly, the respiratory tract virome, which includes latent adenoviruses, anelloviruses, and endogenous retroviral elements, is emerging as another component of mucosal immunity. Altered respiratory virome composition has been observed in asthma and chronic obstructive pulmonary disease, which are often associated with inflammation or disease exacerbation [84]. Relatively few studies have yet addressed the effect of the respiratory virome on responses to inhaled vaccines, such as intranasal influenza vaccines. This topic remains an exciting and important area of future research. One could speculate that the presence of certain non-pathogenic viruses in the airways could precondition the mucosal immune system, thereby altering the interferon response and eventually vaccine performance.

In conclusion, the human virome, ranging from primarily well-known viruses like CMV and hepatitis C to obscure agents like pegiviruses or bacteriophages, represents a rich and largely untapped dimension for immune regulation. Recent advances in metagenomic sequencing have revealed new viruses, such as redondoviruses from the respiratory tract [9], which may provide further knowledge insight into how interindividual variability in virome composition contributes to the polymorphism of vaccine responses. Integrative modeling, which combines virome profiling with immune phenotyping, may provide insight into why individuals with similar demographic and clinical characteristics respond differently to the same vaccine.

## 7. Bacteriophage-Based Vaccines and Their Immunogenic Potential

One field that is constantly progressing in personalized medicine is phageomics, which is concerned with the study of bacteriophage genetic material. Phageomics holds promise for addressing the global issue of antimicrobial resistance by providing easier alternatives to antibiotics [86].

As phages also have the potential to be used use in vaccination, there is growing interest among researchers in their possible benefits. Similar to the key objectives of effective vaccine strategies, phages can induce both cell-mediated and humoral immune responses.

Phage-based vaccines use bacteriophages as delivery vehicles to target the immune system with antigens, aiming to initiate an immune response that protects against a particular disease [87]. Currently, there are three main categories of bacteriophage-based vaccines: display vaccines, DNA vaccines, and hybrid ones [88]. In display vaccines, phages may genetically express proteins or protein fragments, such as peptides, on the coat proteins. Otherwise, antigens may be linked to the surface of the phage via chemical attachment [89].

The major drawbacks of DNA vaccines, such as poor immunogenicity and dependence on adjuvants, can be overcome using bacteriophage-based DNA vaccines. This can be achieved by providing delivery vectors [90]. Phage particles act as strong, suitable carriers for effective DNA delivery, as well as potent adjuvants. These bacteriophage-DNA vaccines have a eukaryotic expression cassette that encodes specific antigens under the control of a target-specific promoter [91].

The most common lambda phages are designed for filamentous phage application [92]. The main advantage of filamentous phage-DNA vaccines is that they can incorporate multiple gene copies at a time, enabling immune activation concerning different antigens with a single delivery vector [92]. In comparison with other vaccines, DNA vaccines, usually incorporated through bacteriophages, are easily produced, relatively inexpensive, exceptionally safe and scalable, and exhibit higher stability, being tolerant to enzymatic degradation, extreme temperatures, and pH gradients [93]. Furthermore, DNA fragments may have a large size, up to 20 kb in the case of lambda phages, and can initiate specific immune responses by directly introducing antigens to immune cells [94].

Considering the unique property of single-stranded RNA bacteriophages to act as antigen carriers, RNA phages have emerged as a candidate at the frontier of vaccine design. Due to their simple genome and rapid replication within their bacterial hosts, phages can be manipulated to hold RNA sequences that encode for specific antigens. The use of RNA phages in vaccine formulation enables the harnessing of both the rapid adaptability of RNA-based technologies and the intrinsic immunostimulatory properties of phages. Phages effectively present poorly immunogenic epitopes, generating a broad immune response that incorporates both humoral and cellular immunity [95]. Recent studies suggest that RNA phage-based vaccines may be excellent and safe candidates for fighting against emerging pathogens and contributing to the ever-expanding armamentarium of modern vaccine strategies [96,97].

Moreover, phages have intrinsic adjuvant activity, allowing them to elicit innate immunity. They can activate TLRs and other PRRs, which are essential for antigen presentation and cytokine production. The success of phage-based vaccines relies on this adjuvant effect. Several studies highlight the potential of phages as adjuvants for vaccines [98,99].

Phage-based vaccines have an outstanding intrinsic advantage: they do not infect human cells and, therefore, cannot cause disease or dangerous cell transformations [55]. Furthermore, these extremely stabilized viral particles can withstand changes in temperature, pH, and chemicals, making them ideal for oral administration and for use in low-resource settings, where maintaining a cold chain is difficult [88].

Another advantage is their ability to activate both humoral and cellular immunity, often without the need of adjuvants. Some studies have shown that phages can amplify immune responses by promoting the release of cytokines and T-cell activation [100]. In addition, phage-based vaccine production is cost-effective and scalable, as the production cost per dose is lower than that of recombinant protein vaccines [101].

However, there are several downsides as well. One common issue is the difficulty in properly displaying antigens at the phage surface, especially when the antigens are large or complex proteins. This decreases the immunological efficacy of the vaccine. Strategies used to overcome this issue include chemical conjugation or dual-display vectors [102].

Additionally, lipopolysaccharide contaminants associated with endotoxins, which are residues from bacterial production, must be sufficiently removed to prevent adverse reactions. Further preclinical studies are needed to understand how lytic phages affect gut microbiota when taken orally [103]. In addition, one important drawback to phage-based vaccines is the potential for horizontal gene transfer, especially that of antibiotic resistance genes, thereby raising biosafety concerns and leading to the enforcement of stringent regulations on genomic engineering and control during phage vaccine development [104].

Despite promising results in animal models and early clinical phases, these vaccines must overcome many regulatory obstacles before they gain approval for use and can be applied on a large scale. Currently, indeed, there are very few large clinical trials underway to demonstrate the efficacy, safety, and industrial standardization of such vaccines.

Considering their fantastic preclinical efficacies and versatility for use in humans, phage-based vaccines are still in the experimental phase. Only a few studies have reached early-phase trials with human patients, while the majority remain preclinical, primarily on animal models. Nevertheless, the different applications and demands for different vaccine platforms continue to advance research in multiple areas.

For example, a phage-based vaccine has been developed against the zoonotic bacterium Brucella abortus. This filamentous M13 bacteriophage was engineered to express Brucella antigens and was shown to induce a Th1-type immune response in murine models [105]. Similarly, phage display technology has been used to design influenza-specific vaccines, in which filamentous phages carrying viral epitopes have induced both cellular and humoral immunity in vivo [106].

Recently, however, research has also focused on using phages against SARS-CoV-2. An experimental RNA-phage platform delivering engineered phages presenting spike protein epitopes was proposed and showed promising immunogenic responses during initial animal trials of testing [107]. While none of these vaccines have moved into advanced human trials, they demonstrate the potential of phages in addressing novel viral threats.

In addition to infectious diseases, phage-based cancer vaccines are also regarded as an emerging area of interest. For instance, M13 phages displaying tumor-specific neoantigens have been assessed in mouse melanoma models, where they promoted both antigen-specific T-cell responses and tumor regression [108]. These results position phages as useful tools for controlling infectious diseases and as candidates for personalized immunotherapy against cancer.

These studies demonstrate the advantages of phage vaccines in terms of production and safety, including very high stability, low production costs, and replication failure in human cells. Nevertheless, challenges remain in standardizing manufacturing processes, eliminating endotoxins, and meeting regulatory standard requirements for clinical approval.

Table 3 outlines representative recent studies that have explored phage-based vaccine applications with various targets and phases of development.

## 8. Clinical Implications: Transplant and Immunocompromised Patients

It is especially critical to understand the impact of the virome on vaccine efficacy, particularly for populations with compromised immune systems. Transplant patients and immunocompromised hosts frequently exhibit suboptimal vaccine responses and thus represent a vulnerable population with a high need for protection. In these subjects, the impact of the virome is intensified, resulting in significant clinical manifestation.

Solid-organ transplant patients are maintained on immunosuppressive medications in order to prevent graft rejection, which in turn suppresses their adaptive immune responses. As is well documented, recipients of organ transplant exhibit substandard responses to conventional vaccinations [109]. For instance, following administration of two doses of the mRNA vaccine for SARS-CoV-2 virus, the majority of kidney or lung transplant patients fail to achieve adequate antibody titers. This finding has led to the administration of additional doses or the implementation of alternative strategies. In this instance, the virome functions as both as a monitor and a modifier of immunity. TTV has emerged as a practical tool in transplant medicine to gauge the net state of immunosuppression. A high TTV load indicates an over-suppressed immune system, whereas a very low load might indicate under-immunosuppression [61]. This same principle applies to vaccine responses: clinicians can potentially use TTV levels to predict which transplant patients are unlikely to respond to a vaccine [61]. Research consortia are now incorporating TTV monitoring in post-transplant vaccine studies (e.g., flu and SARS-CoV-2 vaccines) to stratify patients by immune function. In clinical practice, if a patient receiving a lung transplant has a TTV titer greater than 10^6 copies/mL, it is reasonable to expect a diminished response to vaccine. In such cases it may be advisable to consider proactive measures, such as administering additional booster doses, utilizing more potent adjuvanted vaccines, or even temporarily reducing immunosuppression around the time of vaccination, with the aim to enhance its efficacy. Another virome concern in transplants is CMV, which is a well-known pathogen in that setting. CMV reactivation can cause severe disease in organ recipients and contributes to graft complications [110]. Transplant centers routinely employ antiviral prophylaxis or preemptive therapy for CMV. By controlling CMV, it is possible to achieve two primary benefits: the reduction of direct CMV disease and the elimination of its immunomodulatory effects, which impede vaccine responses. Indeed, studies in transplant patients have shown that those with more effectively controlled CMV, whether through pharmaceutical intervention such as valganciclovir or through adoptive T-cell immunotherapy in stem cell transplants, exhibit enhanced responses to influenza and SARS-CoV-2 vaccines. However, formal clinical trials are necessary to substantiate these findings [111]. In essence, for transplant patients, the management of the virome is becoming part of comprehensive care. This involves the utilization of diagnostic tools such as TTV PCR to assess the virome, along with the implementation of therapeutic interventions like CMV prophylaxis and BK virus monitoring to regulate it. The efficacy of vaccines in this high-risk population is significantly influenced by these strategies [112].

Immunocompromised hosts exhibit a wide spectrum of health conditions, ranging from subjects undergoing chemotherapy or biologic immunosuppressant treatment, to those afflicted with primary immunodeficiencies, and even PLWH, among others. Each of these scenarios presents a unique set of virome interactions. For instance, patients with certain primary immune deficiencies are susceptible to chronic norovirus or enterovirus infections. These ongoing infections can lead to immune exhaustion and inflammation, which may impair the body’s ability to responds to new vaccines. In such cases, some live vaccines are contraindicated entirely. In oncology patients receiving chemotherapy, reactivation of herpesviruses (CMV, varicella zoster virus) is prevalent and can lead to complications. Clinicians sometimes administer prophylactic acyclovir/valacyclovir which, intriguingly, may also indirectly enhance these patients’ capacity to respond to vaccines administered during chemotherapy. This phenomenon, although not extensively studied, shares a conceptual similarity with the vasculitis trial previously mentioned. In subjects with HIV, the primary treatment is antiretroviral therapy to control HIV virus. However, even patients who have been treated for HIV often experience residual immune dysfunction and an expanded virome. According to the findings of studies such as that of Royston et al., the presence of untreated HIV has been shown to result in virome expansion [68]. Higher titers of various gut viruses correlated with immune activation. Despite undergoing treatment, individuals with HIV+ often exhibit elevated TTV levels in comparison to HIV-negative controls, reflecting incomplete immune recovery. This phenomenon can result in diminished vaccine responses, as evidenced by the observation that some PLWH require additional doses of the HBV vaccine to seroconvert, and their antibody titers following SARS-CoV-2 vaccine have been documented to be lower on average (especially in cases where CD4 counts have not been fully restored) [68]. For these patients, additional steps, such as measuring cytomegalovirus IgG or TTV, might refine our understanding of who is truly immunologically “fit” for vaccination versus who might require boosting. Additionally, there is growing interest in the potential of therapeutic interventions targeting the virome to enhance vaccine efficacy in immunocompromised patients. In addition to the administration of antiviral drugs, passive or active immunization against virome components themselves could be considered. For instance, the provision of CMV-specific immunoglobulin or, upon its development, a CMV vaccine to an HIV-positive individual might alleviate the CMV-driven immunosenescence and enhance their responses to other vaccines.

In clinical practice, these insights are leading to substantial changes. Vaccine guidelines for immunocompromised patients have evolved to acknowledge the heterogeneity of this population of patients, recognizing that a universal approach to vaccination is not applicable in this group. For instance, individuals undergoing organ transplant currently receive an intensified for SARS-CoV-2 virus. This entails the administration of three primary doses of the vaccine, instead of the previous two, in addition to the administration of booster shots. The rationale for this adjustment is that their virome and the status of their immunosuppression have been demonstrated to hinder the response to the vaccine [61]. Several medical centers have initiated a pilot program that utilizes immune monitoring, including the assessment of virome markers, to determine the optimal timing of vaccination. This approach involves the deferral of non-urgent vaccinations until a patient’s TTV declines below a threshold following induction of immunosuppression. Conversely, TTV may be monitored and, if necessary, re-vaccinated once it reaches a higher level. For HIV patients, current recommendations frequently recommend revaccination for certain diseases, such as hepatitis B, if initial responses are deemed inadequate. Furthermore, monitoring CMV status may contribute to refining these recommendations in the future.

Overall, acknowledging the virome as both an ally and adversary in the field of vaccinology can assist clinicians better in more effectively protect immunocompromised individuals. By monitoring silent viral partners such as TTV or CMV, we gain insights into the immune readiness of our patients. By mitigating the negative impacts of these viruses through the administration of antivirals, the modification of vaccine formulations, or the strategic timing of vaccinations, it is possible to improve the efficacy of vaccine-based interventions. This is crucial for individuals who are most reliant on vaccines to prevent illness.

A substantial proportion of the knowledge on the role of the virome in immune modulation comes from experimental studies employing animal models and in vitro systems, which allow the investigation of cause-and-effect relationships under controlled conditions.

Mice provide a suitable model system to investigate the impact of chronic viral infections on immune responses to vaccination. Pioneering work using murine CMV (MCMV) infection in mice demonstrated parallels to human CMV: MCMV establishes life-long infection and causes a significant expansion of certain T-cell populations in the mouse. Researchers have shown that MCMV-infected mice exhibit impaired immune responses to new vaccines or pathogens introduced subsequently. For instance, Cicin-Sain et al. [75] infected young adult mice with MCMV and subsequently immunized them with influenza or challenged them with West Nile virus. The MCMV-positive mice exhibited weaker CD8^+^ T-cell responses to the new antigens and had narrower T-cell receptor diversity compared to MCMV-negative controls [75]. This finding suggests a direct correlation between the presence of a persistent virome member and the subsequent suppression of immune responses. The hypothesis proposes that this suppression is due to the “occupation” of immune resources, leading to premature aging of the T-cell pool. Another study conducted on mice co-infected with multiple chronic agents (MCMV, a murine γ-herpesvirus, helminths, etc.) found that such chronic immune activation significantly reduced the antibody response to a live attenuated Yellow Fever 17D vaccine (YF-17D) given to the mice [113]. This phenomenon is analogous to real-world scenarios in which humans often exhibit multiple chronic infections concurrently. The simultaneous presence of these infections can diminish the efficacy of vaccines, as evidenced by the controlled murine experiment.

Conversely, mouse models have demonstrated the beneficial aspects of a virome. The murine norovirus (MNoV) model is instructive in this regard. MNoV can establish a persistent infection in the gastrointestinal tract of mice, which is predominantly asymptomatic, particularly in immunocompetent hosts. In germ-free mice, which are deficient in microbes, the immune system is underdeveloped. These mice have reduced gut lymphocyte numbers and low IgA levels, among other immune system deficiencies. Interestingly, colonization of germ-free mice with an MNoV strain restored many immune deficiencies, thereby compensating for the absence of bacteria [114]. The virus triggered type I interferon-dependent pathways that promoted intestinal lymphocyte function without causing disease [114]. This finding suggests that the virome may contribute to immune system homeostasis. Another classic model is the latent γ-herpesvirus 68 in mice (a model for EBV). Barton et al. employed this model to demonstrate that latency can increase basal levels of interferon-γ and macrophage activation [18]. The latent virus did not cause harm to the mouse; however, when subsequently exposed to a lethal bacterial infection (Listeria or Yersinia pestis), previously infected mice had a survival advantage, attributable to their heightened innate immunity [16,18]. This finding suggests that latent herpes can protect against some infections. However, in the context of vaccination, the chronic IFN-γ and macrophage activation may present a multifaceted phenomenon. On the one hand, it may function as an adjuvant, enhancing immune responses. On the other hand, excessive activation can hinder the generation of a finely tuned adaptive response by inducing immune fatigue or suppressive feedback, thereby compromising immune system function.

Humanized mice are a type of mouse model in which human immune cells or tissues are engrafted, thereby enabling the study of certain human viruses in vivo. For instance, humanized mice have been shown to support latent EBV or CMV infection [115]. Researchers have employed such models to examine the impact of human latent viruses on the development of human immune cells. A study showed that humanized mice latently infected with EBV exhibited alterations in their B- and T-cell repertoires. Furthermore, when vaccinated with a candidate malaria vaccine, these mice exhibited different antibody responses compared to uninfected humanized mice [116]. Studies on humanized mice carrying both HIV and CMV may be useful to evaluate how co-infection may alter vaccine responses. Preliminary findings indicate that co-infected mice exhibit impaired vaccine-specific T-cell responses, mirroring observations in human patients [117]. These models are of outstanding value in the testing of intervention strategies. For instance, the treatment of CMV-infected humanized mice with letermovir, a CMV inhibitor, allows assessment of the efficacy of a test vaccine. This approach provides substantial evidence in support of the implementation of analogous interventions in human subjects [118].

Co-culture experiments allow the dissection of molecular mechanisms. For example, in order to comprehend TTV’s immunomodulation, researchers have incorporated synthetic TTV DNA into cultures of human peripheral blood mononuclear cells and observed the occurrence of TLR9-dependent induction of interferon-alpha. The TTV ORF2 gene has been found to have a significant impact on NF-κB signaling in a human cell line, as confirmed through the use of transfected cells [49]. Such reductionist approaches confirm the ability of viral components to modify cytokine profiles. A similar approach can be used to compare dendritic cells from CMV-seropositive versus CMV-seronegative individuals in vitro. It has been observed that monocyte-derived dendritic cells from CMV+ donors frequently exhibit differential levels of IL-6 and IL-10 upon stimulation. This variation may be attributed to the in vivo exposure of these cells to CMV, which could have led to a process of trained or tolerized response [119]. Another intriguing in vitro system involves the use of immune cell exhaustion assays. For instance, T cells specific to a vaccine antigen (e.g., tetanus toxoid) can be examined in the presence or absence of CMV antigen stimulation [120]. Investigators found that concurrent stimulation with an unrelated persistent virus antigen (such as a CMV peptide) can sometimes divert memory T cells or cause them to express more PD-1, an exhaustion marker, thereby reducing proliferation in response to the vaccine antigen. This provides a mechanistic indication of “distraction”, whereby the immune system’s focus on latent viruses can become diluted when it is concurrently engaged with vaccine components.

In summary, experimental models provide substantial support for clinical observations and offer a mechanistic basis for understanding the impact of chronic viruses on vaccine responses. These models demonstrate that chronic viruses can impede vaccine responses by reducing naive cell pools, causing exhaustion, or altering innate signals. Conversely, certain commensal viruses have been observed to support immunity by providing tonic stimulation that function as an adjuvant. The challenge lies in tearing apart these effects and the subsequent determination of how to accentuate the positive, while mitigating the negative in a real-world setting.

## 9. Future Perspectives: Virome-Aware Precision Vaccinology

As vaccine science advances towards precision immunization—where schedules and formulations are tailored to the unique biology of each person—the integration of an individual’s virome into this paradigm represents a compelling frontier. In the future, vaccine protocols and viromic profiling, which is analogous to host genotyping or microbiome sequencing, may become a routine part of pre-vaccination assessment. A patient’s plasma TTV load can be measured in conjunction with their serostatus for latent viruses, such as CMV and EBV. Those with a “noisy” virome—defined as elevated TTV levels or a positive CMV test result—may be identified as having a high likelihood of suboptimal response. Consequently, they may receive a regimen that incorporates enhanced adjuvants, additional booster doses, or vaccination scheduled during periods of reduced viral activity. Conversely, individuals with minimal latent viral load could follow standard vaccine protocols with expected efficacy. The application of virome-derived biomarkers in clinical trial design, such as stratification or subgroup analyses of clinical trials, is a potential avenue for future research. One specific area of interest is the comparison of vaccine efficacy between CMV-positive and CMV-negative participants [121]. A more profound understanding of the mechanisms by which viral components modulate immunity could potentially contribute to the development of next-generation adjuvant design. For instance, TLR9 activation by TTV DNA has been proposed to promote strengthening Th1 polarization. Consequently, mimicking this effect using synthetic CpG motifs could be a viable therapeutic approach for individuals who lack such innate priming. Conversely, in cases where persistent viral inflammation impedes immune activation, the concurrent administration of short-term immunomodulators and IL-10 mimetics or checkpoint inhibitors can be co-administered to temporally recalibrate the immune environment. A more ambitious approach involves virome manipulation. This process utilizes non-replicating viral vectors to “precondition” the immune system by inducing interferons, thereby creating a condition of defense before administering vaccines [122]. The primary concern in this context pertains to the potential inadvertent skewing of responses due to the innate training, as it could inadvertently replicate beneficial commensal viral stimulation.

The development of vaccines targeting chronic virome members themselves, such as CMV, EBV, or even engineered benign phages, holds significant potential. These vaccines could potentially alleviate the immunologically noisy background conditions, thereby enhancing the efficacy of other vaccines. To illustrate, the implementation of childhood vaccination for CMV has the potential to enhance vaccine responses among the elderly by removing a primary driver of immunosenescence. The efficacy of immune recovery in non-related contexts has already been confirmed with curative infections for hepatitis B and C [123].

The integration of big data and machine learning has been identified as a key enabler, facilitating the acceleration of these projects along their respective trajectories. As immunization programs increasingly collect multi-omic data, including virome composition, algorithms may reveal predictive patterns. For instance, a positive correlation has been observed between CMV+EBV+TTV being high with poor CD8 but normal B-cell responses, suggesting the potential for the development of a virome risk score to predict suboptimal vaccine outcomes [124]. Predictive modeling could serve as a guide for clinicians, informing them of the appropriate timing for adjusting supplementation schedules or doses, or for determining the optimal interval before proceeding with vaccination, similar to the approach employed in HIV and transplant medicine, where the wait period is determined by the achievement of viral suppression or the attainment of a TTV load plateau, respectively [125].

Furthermore, vaccination may be exerting an indirect influence on the composition of the virome. The enhancement of antiviral efficacy through excavation would consequently mirror real-time measurements, as opposed to merely evaluating antibody titer, extending beyond that scope to encompass the suppression of chronic viruses such as TTV. Initial reports, such as those describing transitory interferon and NK cell activation following influenza vaccination, are suggestive of this feedback loop. Therefore, virome shifts might provide a new readout of overall antiviral immunity engendered by the vaccine [126].

In summary, the future of vaccinology can be conceptualized as a holistic perspective, wherein everyone’s silent viral co-inhabitants are mapped, monitored, and modulated alongside the delivery of vaccines. This approach, which has been derived from collaborations between vaccinologists, virologists, immunologists, and data scientists, may eventually hold promise in bringing personalized, highly effective vaccines to reality.

## 10. Conclusions

Recently, the influence of the virome on human biology has gained recognition. Not only does it shape the innate immune response but it also adapts immune responses, according to which it is increasingly becoming a fingerprint of the efficacy of vaccines against viruses. Yet, these connections, largely presented as correlative observations, require much further research into causal and possible mechanistic pathways. Even if many experimental models are informative, they leave much to be desired regarding human systems. Others may suggest that the diversity and dynamics of what constitutes someone’s virome vary across population groups. Nevertheless, the integration of virome-related data into vaccinology is expected to open new avenues for personalized and increasingly effective immunization strategies. There is a need for ongoing multidisciplinary research, which is critical for accelerating the translation of these results into viable public health interventions.

## Figures and Tables

**Figure 1 vaccines-13-00895-f001:**
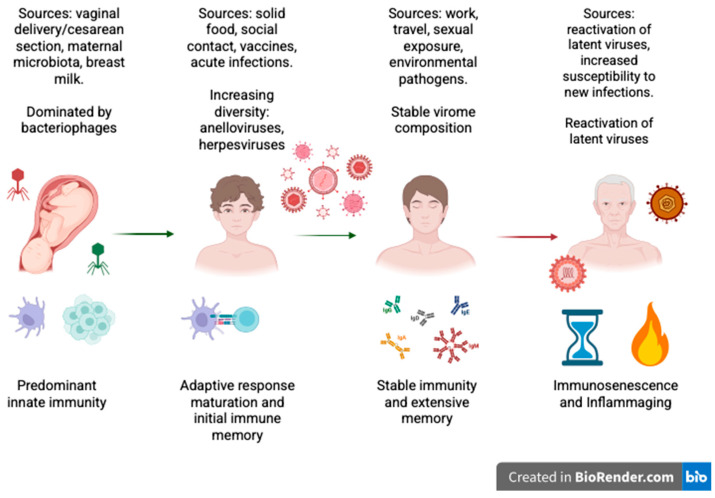
Schematic illustration of how the human virome constantly interacts with the immune system throughout life. The figure illustrates the main sources of viral acquisition during the different stages of life and the dominant immunologic profiles in each phase. In infancy, innate immunity is the primary defense before the acquisition of maternal/environmental viromes. During childhood, virome diversity expands and adaptive immunity develops, leading to the establishment of initial immune memory. In adulthood, the virome reaches a steady state, but it remains immunologically active. Reactivation of latent viruses in old age may take place due to immunosenescence and chronic low-grade inflammation (“inflammaging”). (Created with BioRender.com.)

**Figure 2 vaccines-13-00895-f002:**
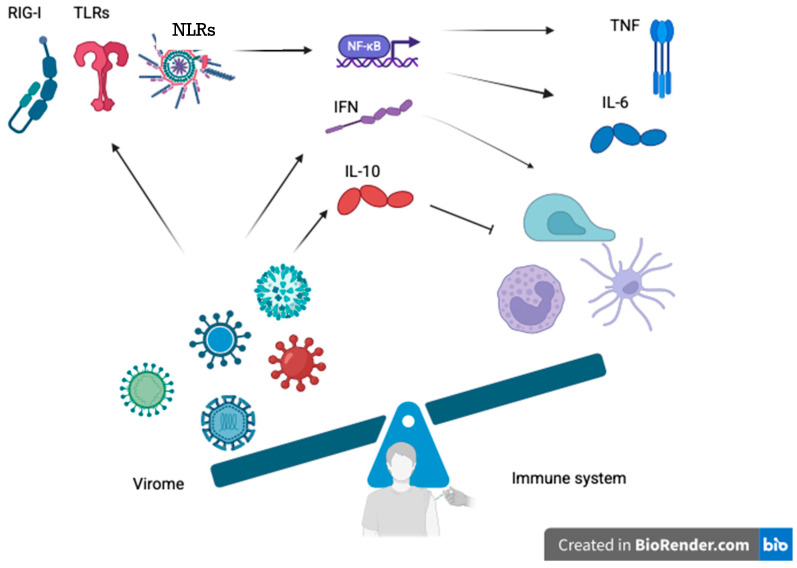
Proposed mechanisms through which the human virome modulates innate and adaptive immunity to vaccines. Persistent or latent viral components (e.g., Torque Teno Virus, cytomegalovirus, noroviruses) activate pattern recognition receptors such as Toll-like receptors (TLR3, TLR7, TLR9), RIG-I-like receptors, and inflammasomes (e.g., NLRP6, NLRP9b). The stimulation causes intracellular signaling cascades to trigger NF-κB activation and production of antiviral interferons (type I and III IFNs), pro-inflammatory cytokines (IL-6, TNF), and regulatory cytokines (IL-10). These immune mediators regulate dendritic cells, natural killers, macrophages, and T and B lymphocytes, thereby shaping vaccine-induced immune responses. The illustration describes the double function of virome-mediated modulation: firstly, in favor of immunogenicity through pre-activation of immune processes (“training” immunity) and, secondly, potentially inducing tolerance, immune exhaustion, or blunted responses. The balance depicted is the dynamic equilibrium between stimulatory and controlling signals coordinated by the virome. Abbreviations: TLR, Toll-like receptor; NLR, NOD-like receptor; NF-κB, nuclear factor kappa-light-chain-enhancer of activated B cells; IFN, interferon; IL, interleukin; TNF, tumor necrosis factor. (Created with BioRender.com.)

**Figure 3 vaccines-13-00895-f003:**
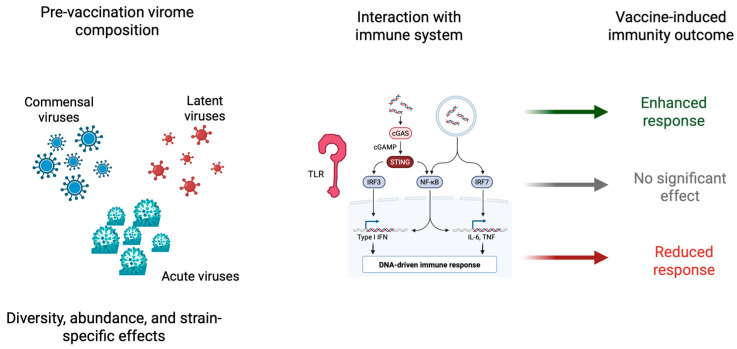
Schematic representation of the speculative role of an individual’s pre-vaccination virome composition in shaping vaccine-induced immune responses. The left panel represents the diversity of virome components—commensal viruses, latent viruses (e.g., herpesviruses, Torque Teno Virus), and acute infections—differing in abundance, diversity, and strain-specific features. The central panel explains the principal intrinsic immune-sensing pathways activated by viral elements, such as Toll-like receptors (TLRs), RIG-I-like receptors, and the cGAS-STING pathway, that converge on intracellular transcription factors IRF3, IRF7, and NF-κB. These pathways trigger the release of type I interferons and inflammatory cytokines. The right panel shows how such pre-existing immune conditions driven by the virome can modulate the vaccines’ immunogenicity to enhance, leave untouched, or diminish responses. The figure highlights the bidirectionality of virome–vaccine interactions: pre-activation may function as a natural adjuvant, while excessive immune noise or tolerance can suppress responsiveness to vaccines. Abbreviations: TLR, Toll-like receptor; NF-κB, nuclear factor kappa-light-chain-enhancer of activated B cells; IRF, interferon regulatory factor; cGAS, cyclic GMP–AMP synthase; STING, stimulator of interferon genes. (Created with BioRender.com.)

**Figure 4 vaccines-13-00895-f004:**
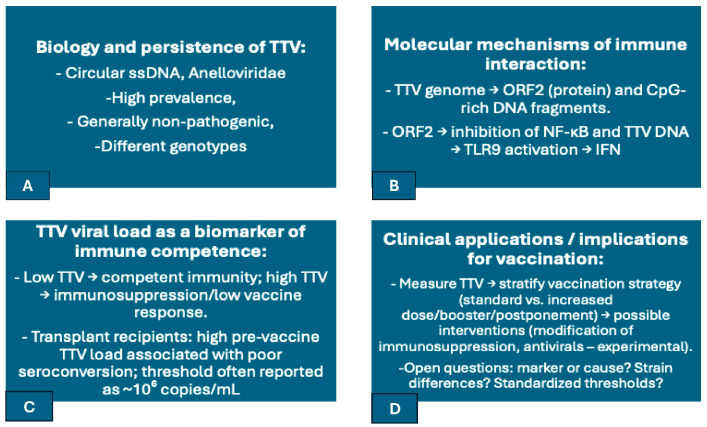
Schematic summary of Torque Teno Virus (TTV) biology, immune modulation mechanisms, and potential implications for vaccine responsiveness. (**A**) TTV is a highly prevalent anellovirus with >90% global prevalence, transmitted early in life and persistent lifelong in blood and lymphoid tissues. (**B**) Mechanistic interactions include NF-κB signaling suppression (by ORF2 protein), CpG motif-TLR9-mediated type I interferon induction, and antiviral pathway modulation by TTV-encoded microRNAs. (**C**) TTV load serves as an immunocompetence surrogate marker, with elevated plasma titers found to be linked to compromised vaccine response in immunosuppressed persons (e.g., transplant recipients, HIV-infected patients). (**D**) Clinical uses include the application of TTV levels to stratify individuals for customized vaccine regimens, to define booster intervals directly, or to optimize immunosuppressive therapy. This dual role of TTV as an immune modulator and biomarker illustrates the regulation that commensal components of the virome exert on vaccine effectiveness. Abbreviations: NF-κB, nuclear factor kappa-light-chain-enhancer of activated B cells; TLR9, Toll-like receptor 9; HIV, human immunodeficiency virus.

**Figure 5 vaccines-13-00895-f005:**
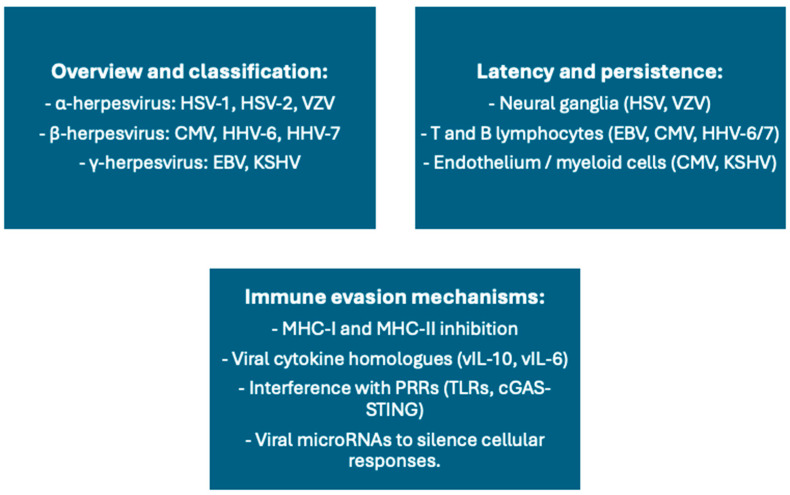
Summary of latent human herpesviruses (α, β, and γ subfamilies), their main sites of latency, immune evasion mechanisms, and potential implications for vaccination. Cytomegalovirus (CMV) is latent in endothelial cells and monocytes, inducing chronic T-cell activation, expansion of senescent T-cell clones, and low-grade inflammation, which can impair vaccine responses, particularly in immunocompromised and aged subjects. Epstein–Barr virus (EBV) infects B cells latently, regulating humoral immunity and possibly suppressing vaccine-induced memory. Human herpesvirus 6 (HHV-6) is latent in lymphocytes and neural tissues, with emerging evidence of effects on HBV and polio vaccine responses. The principal immune evasion strategies include MHC molecule downregulation, blocking interferon signaling, and promotion of exhausted or terminally differentiated lymphocyte populations. The illustration points to the two-faced nature of herpesviruses: in some situations, they act as immune adjuvants through chronic stimulation, but more often as inducers of immunosenescence and reduced responsiveness to vaccine. Abbreviations: CMV, cytomegalovirus; EBV, Epstein–Barr virus; HHV-6, human herpesvirus 6; HBV, hepatitis B virus; MHC, major histocompatibility complex.

**Table 1 vaccines-13-00895-t001:** Viral families in the human virome are classified by genomic type and their main anatomical locations of persistence or replication. They are also characterized as either commensal or pathogenic. The table highlights that some virome members can exhibit lifelong persistence in certain niches such as the bloodstream, lymphoid organs, neuronal or renal cells, and the gastrointestinal tract. In non-pathogenic situations, these viruses may help promote immune homeostasis. However, they can induce pathogenic effects in immunocompromised hosts. For example, Anelloviridae are predominantly non-pathogenic and may serve as markers for immune competence. In contrast, Herpesviridae and Polyomaviridae establish latency and may cause disease when the immune system is suppressed. Bacteriophages do not infect human cells directly, but they influence immunity by controlling the activity and organization of the bacterial microbiota. Abbreviations: ssDNA, single-stranded DNA; dsDNA, double-stranded DNA; CNS, central nervous system. References are to the key studies mentioned in the text.

Virus Family	Genome	Site of Replication/Persistence	Commensal or Pathogenic	Reference
Anelloviridae	Circular ssDNA	Blood, lymphoid tissues; lifelong persistence	Commensal (non-pathogenic)	[7]
Herpesviridae	Linear dsDNA	Latent in neurons, epithelial cells, lymphocytes	Pathogenic or immunomodulatory	[8]
Polyomaviridae	Circular dsDNA	Kidney, urinary tract, CNS	Mostly commensal; opportunistic	[11]
Papillomaviridae	Circular dsDNA	Skin and mucosal epithelia	Both commensal and oncogenic types	[11]
Parvoviridae	Linear ssDNA	Blood, bone marrow	Potentially pathogenic	[11]
Redondoviridae	Circular ssDNA	Oropharyngeal and respiratory tract	Commensal (elevated in disease)	[5]
CrAss-like phages	Linear dsDNA	Gut mucosa and lumen; infect *Bacteroidota* spp.	Commensal (dominant phage family)	[4]
Microviridae, Siphoviridae, etc.	ssDNA or dsDNA (phages)	Gut; infect gut-resident bacteria	Commensal	[4]
Endogenous retroviruses	Integrated DNA	Human genome (germline-encoded)	Mostly inactive/commensal	[11]

**Table 2 vaccines-13-00895-t002:** Summary of the most relevant human virome members and their probable function in modulating vaccine-induced immunity. The table summarizes the viral family, genomic features, epidemiological prevalence, and characterized or hypothetical functions in vaccine response-related immunomodulation. Torque Teno Virus (TTV) is characterized as a marker of immune competence surrogate, with high viral load associated with poor responses to vaccination in transplant recipients and HIV-infected patients. Herpesviridae members such as CMV and EBV, by lifelong latency and immune remodeling, can alter T- and B-cell dynamics and therefore affect vaccine efficacy. Polyomaviruses (BK/JC) are indicators of immunosuppression under transplantation, while pegiviruses (GBV-C) could be protective immunomodulators in co-infected individuals. Noroviruses and γ-herpesviruses in mouse models illustrate the pre-conditioning of immune responses by chronic viral infections to boost or limit vaccine efficacy. Bacteriophages are highlighted as indirect controllers, modulating vaccine responses through microbiome modulation and metabolite secretion. This table highlights the virome–vaccine interaction complexity and the importance of integrating virome profiling into personalized vaccinology. Abbreviations: dsDNA, double-stranded DNA; ssRNA, single-stranded RNA; TTV, Torque Teno Virus; CMV, cytomegalovirus; EBV, Epstein–Barr virus; HBV, Hepatitis B virus.

Virus	Family/Genus	Key Characteristics	Role in Vaccine Immunity	Notes/References
TTV	*Anelloviridae* Alphatorquevirus	Circular ssDNA; ubiquitous, non-pathogenic	Inverse marker of immune competence; high TTV load correlates with poor vaccine response in transplant and HIV+ patients	Clinically relevant; extensively cited [30,31,33,34]
CMV	*Herpesviridae* Betaherpesvirus	dsDNA latency; lifelong infection	Alters vaccine responses, particularly in the elderly and immunocompromised; can impair CD8+ responses and T-cell repertoire	CMV seropositivity reduces vaccine efficacy [42,43,44,45,46,47],
EBV	*Herpesviridae* Gammaherpesvirus	Latent in B cells; widespread	Modulates B-cell dynamics; potential impact on long-term humoral memory	Suggested in B-cell vaccination studies [44,48,49]
HHV-6	*Herpesviridae* Betaherpesvirus	Prevalent latent virus; neurotropic potential	Possible impact on vaccine efficacy (e.g., HBV, polio); underexplored	Mechanistic roles unclear [49,50]
BK/JC	*Polyomaviridae*	dsDNA; latent in the urinary tract	Used as a surrogate marker of immunosuppression in transplant recipients	Potential biomarker in immunized hosts [43]
GB Virus C (Pegivirus)	*Flaviviridae* Pegivirus	ssRNA; non-pathogenic, persistent in blood	Modulates T-cell activation and cytokine balance; protective in HIV co-infection	Immunomodulatory in co-infection models [51,52]
Norovirus (murine/human)	*Caliciviridae*	ssRNA; enteric, persistent in mice and humans	Promotes intestinal interferon signaling; model for virome–immune homeostasis	Shown in murine vaccine priming [53,54]
Murine CMV/γHV-68	*Herpesviridae*	Mouse models of latent herpesvirus infection	Boosts baseline IFN-γ and antimicrobial resistance; impacts vaccine recall responses	Experimental models of latent infection [5,6,8,55]
Bacteriophages	Various	Highly abundant on mucosal surfaces; infect commensal bacteria	Indirectly modulates the immune response via microbiome shaping; alters vaccine response through metabolite and barrier effects	Key component of mucosal virome [22,25,56,57]

**Table 3 vaccines-13-00895-t003:** Representative examples of bacteriophage-based vaccine platforms, targets, phage type, development status, and primary outcomes. The table demonstrates the versatility of phages as antigen carriers for use against infectious diseases (e.g., Brucella abortus, influenza, SARS-CoV-2) and cancer immunotherapy (neoantigen-presenting phages). Display vaccines rely on the expression or chemical conjugation of antigenic peptides on phage surfaces, while DNA and RNA phage vaccines are delivery vectors with intrinsic potent adjuvant function. Examples include M13 phages expressing Brucella antigens, eliciting Th1 responses in mice; filamentous phages as vectors for influenza DNA vaccines; RNA phages bearing SARS-CoV-2 spike epitopes; and phage display vaccines against melanoma neoantigens. These early and preclinical studies demonstrate the theoretical advantages of phage vaccines—cost, stability, capacity to induce humoral and cellular immunity, and safety due to inability to replicate in human cells—while identifying key limitations with regard to endotoxin contamination, horizontal gene transfer risks, and manufacturing standardization. *Abbreviations:* Th1, T-helper 1 cells; SARS-CoV-2, severe acute respiratory syndrome coronavirus 2; RNA, ribonucleic acid; DNA, deoxyribonucleic acid. References match the studies cited in the text.

Study/Project	Target	Phage Type	Status	Notes	References
Phage-based vaccine for *Brucella abortus*	Brucellosis (zoonosis)	M13	Phase I (animal model)	Induces Th1 immune response	[105]
Bacteriophage as DNA vaccine vector	Influenza	Filamentous (fd)	Phase I (mouse model)	Elicits strong immunogenicity	[106]
Phage display vaccine for SARS-CoV-2	COVID-19	RNA phage + spike epitopes	Preclinical	Is immunogenic in mice	[107]
Phage-based cancer vaccine (neoantigen-M13)	Melanoma/solid tumors	Modified M13	Preclinical (animals)	Induces anti-tumor T-cell response	[108]

## Data Availability

No new data were created or analyzed in this study.

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
