# Peer review of "Role of the Virome in Vaccine-Induced Immunization"

_vaccines, 2025, doi:10.3390/vaccines13090895_

Round 1
Reviewer 1 Report
Comments and Suggestions for Authors
The review article submitted by Rossella Cianci and coauthors is a balanced review article and written in good form. As the pictures can improve the understandings of the subject, I would recommend adding 1-2 other figures to improve the quality of the review article. Figures can explain how virom in humans develops and how it can interact with different immune factors. Also how virom can impact the immunization of different vaccines.
Also add the different functions and roles of the virome in the host, such as what is the effect on the microbiome, some non-inflammatory diseases, etc.—a brief part of the functions and roles of the virome.
Conclusions need to be in well balanced on the basis of preivously published data.
Author Response
Dear Editor of Vaccines
First, my coauthors and I would like to thank you sincerely for this opportunity to cooperate. We profoundly thank the reviewers for the comments and useful suggestions to improve the paper. We thank You for your constructive critique and hope the review process has improved the manuscript. If additional changes are warranted, we will make them. 
We hope that this revised version of our manuscript may now be found suitable for publication. 
This is a point-by-point list of changes made in the paper:
REVIEWER 1
- The review article submitted by Rossella Cianci and coauthors is a balanced review article and written in good form. As the pictures can improve the understandings of the subject, I would recommend adding 1-2 other figures to improve the quality of the review article. Figures can explain how virom in humans develops and how it can interact with different immune factors. Also how virom can impact the immunization of different vaccines.
We have added two new figures, as suggested.
- Also add the different functions and roles of the virome in the host, such as what is the effect on the microbiome, some non-inflammatory diseases, etc.—a brief part of the functions and roles of the virome.
We have included several passages in the manuscript describing the functions and roles of the virome in the host, including interactions with the bacterial microbiome and associations with non-inflammatory diseases (e.g., type 2 diabetes, obesity, neurodevelopmental disorders, certain cancers) [6,24,25]. These concepts are present in the introductory section, in the part on immune modulation, and in the discussion on the role of bacteriophages.
- Conclusions need to be in well balanced on the basis of preivously published data.
We have rewritten the conclusions, as suggested.
We thank You for your constructive critique and we hope the review process has led to an improved manuscript.
If additional changes are warranted, we will make them.
We hope that this revised version of our manuscript may now be found suitable for publication.
Sincerely,
Rossella Cianci on behalf of co-authors.
Reviewer 2 Report
Comments and Suggestions for Authors
The review article submitted to Vaccines, entitled “Role of the Virome in vaccine induced immunization,” addresses a very interesting and relevant topic. However, I recommend that the authors present the content in a more concise manner, supported by clear and well-designed schematics, to better engage the reader and enhance the overall impact. At present, the points raised do not come across as firmly or memorably as they could. I encourage the authors to be more specific and precise in their arguments and explanations. Please find below some detailed suggestions.
Section 2: The human virome and immune system
(i) It is important to establish the term virome and sufficient evidence should be given. Hence give a brief summary of term Microbiome, how it derived and how it revolutionized the field of immunology. Then focus on VIROME and give a summary of different families of viruses that can be considered to exist in virome and classifying them either to commensal or pathogen. Make it different from the Table which is already included.
(a) Include viruses and virus families (Do extensive search)
(b) Sites of their replication and sites of persistence
(c) Commensal or pathogen
(d) References
(ii) What if the acute viral infection results in persistence and whether it will add to the virome and whether it will affect the immune system (Good to think about it and mention viral persistence a brief after chronic/acute infection).
Section 3: Role of virome in vaccine-induced immunization
Basically this section adds more to the Section 2. Not much is mentioning about the how the virome is affecting the vaccine-induced immunization, but rather it mentions about the how it modulates general innate and adaptive immune system. Preferably change the title. So may be the authors can restructure the Section 2 and 3
as
(i) Historical setting of microbiome and virome (Table 1 as mentioned above)
(ii) How virome can affect the immune system and provide all evidence mentioned in Section 2 and 3 here. (Figure 1 and Table 2 -the current one)
(iii) Give hypothesis/evidence for how virome may affect the vaccine induced immunization (a short paragraph)
Figure: The current figure is very primitive and doesn’t make any sense.
Include all evidence mentioned in Section 2 and 3 – include all pathways, its outcome both IFN pathway or proinflammatory cytokine production via transcription, which all viruses (in virome) are already shown to modulate this. Be more professional.
Section 4: TTV
This is very interesting and not known much to public. So a good schematic on virus and current evidence will be good to catch the attention for how it might modulate the vaccination response.
Section 5: Latent Herpesviruses
Here also I expect a schematic so that people can always refer to text, but a major information is received from the figure.
Section 6: I find two sections here, so advice to split them – first is commensal viruses and section 7 to be Bacteriophage based vaccines and their response.
Be more concise and specific.
Author Response
Dear Editor of Vaccines
First, my coauthors and I would like to thank you sincerely for this opportunity to cooperate. We profoundly thank the reviewers for the comments and useful suggestions to improve the paper. We thank You for your constructive critique and hope the review process has improved the manuscript. If additional changes are warranted, we will make them. 
We hope that this revised version of our manuscript may now be found suitable for publication. 
This is a point-by-point list of changes made in the paper:
REVIEWER 2
The review article submitted to Vaccines, entitled “Role of the Virome in vaccine induced immunization,” addresses a very interesting and relevant topic. However, I recommend that the authors present the content in a more concise manner, supported by clear and well-designed schematics, to better engage the reader and enhance the overall impact. At present, the points raised do not come across as firmly or memorably as they could. I encourage the authors to be more specific and precise in their arguments and explanations. Please find below some detailed suggestions.
- Section 2: The human virome and immune system
(i) It is important to establish the term virome and sufficient evidence should be given. Hence give a brief summary of term Microbiome, how it derived and how it revolutionized the field of immunology. Then focus on VIROME and give a summary of different families of viruses that can be considered to exist in virome and classifying them either to commensal or pathogen. Make it different from the Table which is already included.
(a) Include viruses and virus families (Do extensive search)
(b) Sites of their replication and sites of persistence
(c) Commensal or pathogen
(d) References
(ii) What if the acute viral infection results in persistence and whether it will add to the virome and whether it will affect the immune system (Good to think about it and mention viral persistence a brief after chronic/acute infection).
We have made the changes to section 2 and created a new table as requested.
- Section 3: Role of virome in vaccine-induced immunization
Basically this section adds more to the Section 2. Not much is mentioning about the how the virome is affecting the vaccine-induced immunization, but rather it mentions about the how it modulates general innate and adaptive immune system. Preferably change the title. So may be the authors can restructure the Section 2 and 3 as
(i) Historical setting of microbiome and virome (Table 1 as mentioned above)
(ii) How virome can affect the immune system and provide all evidence mentioned in Section 2 and 3 here. (Figure 1 and Table 2 -the current one)
(iii) Give hypothesis/evidence for how virome may affect the vaccine induced immunization (a short paragraph)
We have modified section 3, as suggested.
- Figure:The current figure is very primitive and doesn’t make any sense.
Include all evidence mentioned in Section 2 and 3 – include all pathways, its outcome both IFN pathway or proinflammatory cytokine production via transcription, which all viruses (in virome) are already shown to modulate this. Be more professional.
As requested by other reviewers, we have supplemented the text with 2 additional figures.
- Section 4:TTV
This is very interesting and not known much to public. So a good schematic on virus and current evidence will be good to catch the attention for how it might modulate the vaccination response.
We have made the section more fluid and inserted a schematic diagram, as requested.
- Section 5: Latent Herpesviruses
Here also I expect a schematic so that people can always refer to text, but a major information is received from the figure.
We have also included a diagram in this section, as requested.
- Section 6: I find two sections here, so advice to split them – first is commensal viruses and section 7 to be Bacteriophage based vaccines and their response.
We have divided section 6, as requested.
We thank You for your constructive critique and we hope the review process has led to an improved manuscript.
If additional changes are warranted, we will make them.
We hope that this revised version of our manuscript may now be found suitable for publication.
Sincerely,
Rossella Cianci on behalf of co-authors.
Reviewer 3 Report
Comments and Suggestions for Authors
The manuscript “ROLE OF THE VIROME IN VACCINE-INDUCED IMMUN IZATION”, submitted to the journal Vaccines , is devoted to the reviewing the components of the human virome amd its impact both innate and adaptive immune responses, including mechanisms of immune activation, trained immunity, and molecular mimicry from both beneficial and detrimental perspectives for vaccine-induced immunization. The theme is of great importance for the improvement the effectiveness of vaccination against viral infections.
I have some comments:
- Lines 57-59: “Non-pathogenic viruses include small circular DNA anelloviruses such as TTV, and large DNA viruses such as herpesviruses that establish latency”.
Comment:
Herpes viruses that cause diseases in humans and animals, including death, can hardly be called “non-pathogenic”.
- Antiviral protection is provided not only by TLR 9, but also by TLR 3 and TLR 7. In addition, the authors discuss the activation of NLRPs under the influence of viruses. It is advisable to write TLRs instead of TLR 9 in the figure, and also add NLRs to which NLRP belongs.
- Information about “Enterovirus B and cosavirus, found in Ghanaian children who did not respond to the Rotarix vaccine” is duplicated. Correction is needed.
- Lines 414-415:
“Latent viruses often secrete homologs of cytokines or evade innate sensors, modulating the innate immune system.”
Comment:
It would be better to change the wording of the phrase. Viruses cannot secrete interferon homologs.
- For the sake of completeness, it is advisable to indicate the number of infectious diseases caused by viruses, as well as how many of them have approved vaccines.
- The Abstract talks about molecular mimicry, which must be taken into account when vaccinating, but the text does not mention molecular mimicry.
- It should be noted that the limiting factor of bacteriophage-based vaccines is the horizontal transfer of antibiotic resistance genes.
Author Response
Dear Editor of Vaccines
First, my coauthors and I would like to thank you sincerely for this opportunity to cooperate. We profoundly thank the reviewers for the comments and useful suggestions to improve the paper. We thank You for your constructive critique and hope the review process has improved the manuscript. If additional changes are warranted, we will make them. 
We hope that this revised version of our manuscript may now be found suitable for publication. 
This is a point-by-point list of changes made in the paper:
REVIEWER 3
The manuscript “ROLE OF THE VIROME IN VACCINE-INDUCED IMMUN IZATION”, submitted to the journal Vaccines , is devoted to the reviewing the components of the human virome amd its impact both innate and adaptive immune responses, including mechanisms of immune activation, trained immunity, and molecular mimicry from both beneficial and detrimental perspectives for vaccine-induced immunization. The theme is of great importance for the improvement the effectiveness of vaccination against viral infections.
I have some comments:
- Lines 57-59: “Non-pathogenic viruses include small circular DNA anelloviruses such as TTV, and large DNA viruses such as herpesviruses that establish latency”.
Comment:
Herpes viruses that cause diseases in humans and animals, including death, can hardly be called “non-pathogenic”.
We have corrected as requested.
- Antiviral protection is provided not only by TLR 9, but also by TLR 3 and TLR 7. In addition, the authors discuss the activation of NLRPs under the influence of viruses. It is advisable to write TLRs instead of TLR 9 in the figure, and also add NLRs to which NLRP belongs.
We have made the changes to the image and caption as requested.
- Information about “Enterovirus B and cosavirus, found in Ghanaian children who did not respond to the Rotarix vaccine” is duplicated. Correction is needed.
We have modified it to eliminate redundancy as requested.
- Lines 414-415:“Latent viruses often secrete homologs of cytokines or evade innate sensors, modulating the innate immune system.
Comment: It would be better to change the wording of the phrase. Viruses cannot secrete interferon homologs.
We have rephrased the sentence correctly.
- For the sake of completeness, it is advisable to indicate the number of infectious diseases caused by viruses, as well as how many of them have approved vaccines.
We have added the information requested in the introduction.
- The Abstract talks about molecular mimicry, which must be taken into account when vaccinating, but the text does not mention molecular mimicry.
We have clarified the concept, as requested.
- It should be noted that the limiting factor of bacteriophage-based vaccines is the horizontal transfer of antibiotic resistance genes.
We have emphasized this concept, as requested.
We thank You for your constructive critique and we hope the review process has led to an improved manuscript.
If additional changes are warranted, we will make them.
We hope that this revised version of our manuscript may now be found suitable for publication.
Sincerely,
Rossella Cianci on behalf of co-authors